# OpenReview forum: "Body Transformer: Leveraging Robot Embodiment for Policy Learning"
_robot-learning.org/CoRL/2024/Conference — CoRL 2024_

### Official Review · Reviewer_tLjG · 2024-07-20
**Body Transformer: Leveraging Robot Embodiment for Policy Learning**

**Originality:** 3
**Technical Quality:** 3
**Clarity Of Presentation:** 3
**Potential Impact:** 3
**Recommendation:** 3
**Confidence:** 4

**Review:**

The paper provides a thorough theoretical foundation for the proposed approach and supports its claims with extensive experimental results. The experiments convincingly demonstrate that BoT outperforms both MLP and vanilla transformer baselines in imitation learning tasks, showing superior performance and generalization capabilities. Additionally, BoT's ability to scale effectively with an increasing number of parameters highlights its robustness and efficiency. In the context of reinforcement learning, BoT exhibits improved sample efficiency and asymptotic performance, particularly with the BoT-Mix variant, which balances local and global information pooling.

However, the reliance on transformer layers may introduce a significant number of trainable parameters, potentially increasing the computational burden. The paper could also benefit from a more detailed discussion on the practical implications of deploying BoT in real-world scenarios, including the challenges of sim-to-real transfer and the potential need for adaptation mechanisms.

**Quality Of The Limitations Section:**

2

**Questions For Rebuttal:**

How does BoT handle the potential computational burden introduced by the additional transformer layers, especially in resource-constrained environments?

Can the authors provide more insights into the practical implications and challenges of deploying BoT in real-world robotic applications?

Could the authors elaborate on the limitations related to the minimum number of transformer layers required to maintain representation power?

**Robotics Focus:**

3

**Summary Of Paper:**

The paper introduces the Body Transformer (BoT), a novel architecture that leverages the physical structure of robots to improve policy learning. By representing the robot's body as a graph of sensors and actuators and using masked attention, BoT integrates the spatial relationships within the robot's body into the learning process. This approach outperforms traditional transformer architectures and multilayer perceptrons in terms of task completion, scalability, and computational efficiency in both imitation and reinforcement learning contexts. The paper includes comprehensive experiments demonstrating BoT's effectiveness and computational advantages.

**Summary Of Recommendation:**

The BoT presents a novel and effective approach to leveraging robot embodiment in policy learning, with significant contributions in both theoretical development and experimental validation.

---

### Official Review · Reviewer_W7Gq · 2024-07-21
**This is an interesting paper that proposes a very simple yet effective idea.**

**Originality:** 4
**Technical Quality:** 4
**Clarity Of Presentation:** 5
**Potential Impact:** 4
**Recommendation:** 3
**Confidence:** 5

**Review:**

## Quality:
### Strengths:
Methodology: The paper employs a comprehensive experimental setup to validate the proposed Body Transformer (BoT) architecture across various tasks, including imitation learning and reinforcement learning.
Detailed analysis: It provides in-depth analysis and comparisons with existing models, demonstrating significant improvements in performance and computational efficiency.
Real-world validation: The real-world experiments on a Unitree A1 robot enhance the practical relevance and applicability of the proposed method.

### Weaknesses:
Implementation details: Some implementation specifics, such as hyperparameters and training details, might be lacking, which can hinder reproducibility.

## Clarity:
### Strengths:
Clear structure: The paper is well-structured, with distinct sections covering background, methodology, experiments, and conclusions.
Figures and Tables: Effective use of figures and tables to illustrate the architecture, experimental setups, and results, aiding in comprehension.

### Weaknesses:
Fewer explanations about mask M: Some sections, especially those explaining the novel masking mechanism, could benefit from more detailed, step-by-step explanations.

## Originality:
### Strengths:
The concept of leveraging the robot's physical structure through a graph representation and integrating it with transformer architecture is innovative.


## Significance:
### Strengths:
Impact on robotics: The proposed architecture has the potential to significantly advance the field of robot learning by improving the efficiency and performance of learning policies.
Broad applicability: The approach is versatile and can be applied to various robotic tasks and configurations, enhancing its significance.

### Weaknesses:
Generalizability: When applying the proposed method to robots with varying physical characteristics (such as humanoid robots with different numbers of joints), it appears essential to tailor the learning process to accommodate each specific robot.

**Quality Of The Limitations Section:**

2

**Questions For Rebuttal:**

## Construction of Mask M:
In this study, the mask M utilized by the proposed model is determined based on the robot's morphology, specifically the arrangement of sensors and actuators. While the experiments conducted in this research employed Unitree A1 and Humanoid in Issac Gym, a detailed explanation of how each sensor and actuator was mapped to the mask M is necessary.
Additionally, it should be noted that sensors on a robot do not only gather information specific to their physical location but may also collect data from distant spatial areas (e.g., cameras or IMUs). Similarly, actuators might operate joints located in different spatial positions. It would be beneficial to discuss how mask M should be constructed in such cases.

## Assumptions on the Sparsity of Mask M:
In the computational analysis, the sparsity of mask M is assumed to be β=0.908, based on the MoCapAct example. However, when the number of sensor and actuator nodes on a robot increases, can this assumption truly hold? When considering a single robot in a detailed graph representation, the sparsity might be reduced. I would appreciate a discussion on this point.

**Robotics Focus:**

4

**Summary Of Paper:**

This paper proposes a new architecture called "Body Transformer (BoT)." BoT represents the robot's body structure as a graph, treating sensors and actuators as nodes. This architecture utilizes a masked attention mechanism, where each node only references information from itself and its adjacent nodes, making robot learning more efficient than traditional transformers. Key contributions include demonstrating that BoT outperforms conventional methods in imitation and reinforcement learning with improved computational efficiency. BoT is shown to be particularly effective for complex tasks and high-dimensional systems, and it has been validated for real-world robot control applications.

**Summary Of Recommendation:**

This paper proposes a new architecture called "Body Transformer (BoT)." BoT represents the robot's body structure as a graph, where each node only references information from itself and its adjacent nodes, making robot learning more efficient than traditional transformers. The proposed method is highly intriguing. I am confident that addressing a few specific points of improvement will significantly enhance the quality of this paper.

---

### Official Review · Reviewer_f458 · 2024-07-22
**novel architecture, insufficient evaluation**

**Originality:** 3
**Technical Quality:** 2
**Clarity Of Presentation:** 3
**Potential Impact:** 2
**Recommendation:** 3
**Confidence:** 4

**Review:**

Strengths:

- The proposed approach is straightforward and easy to follow.
- The approach is validated in both simulation and real-world robots.
- It effectively leverages the sparsity of the graph structure to reduce computation.

Weaknesses:

The main area that needs improvement in my opinion is the evaluation, which is currently insufficient to fully assess the efficacy of the proposed method:

- In Fig.3(a), the “maximum value achieved throughout three trajectories” is reported. Why over only 3 trajectories? Compared to Fig. 7, it seems to be a typo and actually meant “seeds”.  Reporting the maximum value alone doesn’t provide a complete picture of performance. I think that including mean and variance, and also increasing the number of seeds, would strengthen the results.
- The RL results In Fig. 4(a) are based on 3 seeds, which is insufficient. Increasing the number of seeds would enhance the reliability of the result.
- Fig. 3(b) seems to be based on a single seed.

The real-world deployment demonstrates feasibility but lacks details on the success rate or failure scenarios. Providing this information would give a more comprehensive understanding of the practical application of the proposed approach.

**Quality Of The Limitations Section:**

3

**Questions For Rebuttal:**

- The metric on the left side in Fig. 7 is ambiguous, “maximum mean value recorded throughout all evaluation epochs across three seeds”. Can you elaborate?
- Can you provide more details about the real-world deployment, such as success rate, failure scenarios, or any challenges encountered?
- How does the epoch on the x-label in Fig. 4(a) correspond to environment steps?

**Robotics Focus:**

4

**Summary Of Paper:**

The paper proposes a transformer-based architecture, body transformer, which encodes the robot morphology as a graph of sensors and actuators. They demonstrate that body transformer based policy network achieves high performance in both imitation learning and RL setting. Furthermore, its performance scales well with the number of parameters in the imitation learning setting. The paper also highlights that the proposed method reduce the computation by 50% compared to a vanilla transformer, thanks to the efficient implementation of the masked attention layers. Additionally, they achieve zero-shot sim-to-real transfer in quadruped robot locomotion.

**Summary Of Recommendation:**

** Updated post-rebuttal **: The proposed architecture is straightforward and relatively efficient to compute. The clarity of the paper and the quality of the experiments have improved following the rebuttal.

---

### Author Rebuttal · Authors · 2024-08-08

We would like to thank all reviewers for their in-depth analysis and helpful comments. The reviewers have recognized that the Body Transformer architecture is innovative, pointing out the significant gains achieved across our extensive simulated and real-world experiments in imitation and reinforcement learning.

We have addressed **all** remaining concerns, specifically:
* Increasing the number of seeds across all experiments, confirming the original findings
* Providing a simple set of guidelines to build the embodiment mask for each robot in a matter of minutes
* Providing additional details regarding the embodiment mask, including an example for the A1 robot, as well as practical considerations and quantitative results regarding the real-world deployment.

In addition, we have extended our set of imitation learning experiments to a dexterous manipulation scenario, and provided additional ablations as suggested by the reviewers.

The revised manuscript is attached to this rebuttal, with the main changes highlighted in yellow. We are looking forward to an engaging discussion.

---

### Decision · Program_Chairs · 2024-09-04

**Decision:**

Accept

**Comment:**

Strengths:
+ A novel architecture that incorporates robot morphologies for robot learning.
+ Experiments include both simulated and real-robot results.
+ Body Transformer achieves significant gains in performance and efficiency.


Weaknesses:
- The RL results are averaged across a limited number of seeds.
- Body Transformer needs to be tailored to each embodiment, which is non-trivial.
- Some details regarding the mask and real-robot deployment are missing.

Post rebuttal:
The additional experiments and author responses have addressed the reviewers' concerns.